# Seasonal cycle of desert aerosols in West Africa : analysis of the coastal transition with passive and active sensors

Habib Senghor [1], Éric Machu [1,2], Frédéric Hourdin [3], and Amadou Thierno Gaye [1]

[1]Laboratoire de Physique de l'Atmosphère et de l'Océan Siméon-Fongang (LPAO-SF), École Supérieure Polytechnique (ESP) de l'Université Cheikh Anta Diop de Dakar (UCAD), Sénégal.
[2]Laboratoire d'Océanographie Physique et Spatiale (LOPS), IUEM, Université Brest, CNRS, IRD, Ifremer, Brest, France.
[3]Laboratoire de Météorologie Dynamique (LMD), CNRS/IPSL/UMPC, Paris, France.

*Correspondence to:* H. Senghor (habib.senghor@ird.fr)

**Abstract.** The impact of desert aerosols on climate, atmospheric processes and the environment is still debated in the scientific community. The extent of their influence remains to be determined and particularly requires a better understanding of the variability of their distribution. In this work, we studied the variability of these aerosols in West Africa using different types of satellite observations. SeaWiFS (Sea-Viewing Wide Field-of-View Sensor) and OMI (Ozone Monitoring Instrument) data have been used to characterize the spatial distribution of mineral aerosols from their optical and physical properties over the period 2005-2010. In particular, we focused on the variability of the transition between the West African continent and the Eastern Atlantic Ocean. Data provided by the Lidar scrolling CALIOP (Cloud-Aerosol Lidar with Orthogonal Polarization) onboard the satellite CALIPSO (Cloud Aerosol Lidar and Infrared Pathfinder Satellite Observations) for the period 2007-2013 were then used to assess the seasonal variability of the vertical distribution of desert aerosols. We first obtained a good representation of Aerosol Optical Depth (AOD) and Single Scattering Albedo (SSA) by satellites SeaWiFS and OMI respectively in comparison with AERONET estimates, both above the continent and the ocean. Dust occurrence frequency is higher in spring and boreal summer. In spring, the highest occurrences are located between the surface and 3 km above sea level, while in summer the highest occurrences are between 2 and 5 km altitude. The vertical distribution given by CALIOP also highlights an abrupt change at the coast from spring to fall with a layer of desert aerosols confined in an atmospheric layer uplifted from the surface of the ocean. This uplift of the aerosol layer above the ocean contrasts with the winter season during which mineral aerosols are confined in the atmospheric boundary layer. Radiosondes at Dakar Weather Station (17.5°W, 14.74°N) provide basic thermodynamic variables which partially give causal relationship between the layering of the atmospheric circulation over West Africa and their aerosol contents throughout the year. A SSA increase is observed in winter and spring at the transition between the continent and the ocean. The analysis of mean NCEP (National Center for Environmental Prediction) winds at 925 hPa between 2000 and 2012 suggest a significant contribution of coastal sand sources from Mauritania in winter which would increase SSA over the ocean.

## 1 Introduction

The Sahara is the largest source of mineral aerosols in the world, with a contribution of almost 40% compared to the overall emissions from natural sources (Ramanathan et al., 2001; Tanaka et al., 2005). The mineral dust aerosols emitted from the Sahara desert can be transported over long distances in the atmosphere and can be detected as far as Americas (Prospero et al., 1981; Swap et al., 1992; Formenti et al., 2001; Kaufman et al., 2005; Ansmann et al., 2009; Ben-Ami et al., 2010), Mediterranean region (Bergametti et al., 1989; Moulin, 1997; Ansmann et al., 2003) and Asia (Ganor and Mamane, 1982; Israelevich et al., 2003; Ganor et al., 2010). But here, the study of dust transport focuses on the main corridor of their transport Westward of Africa (Formenti et al., 2001). They play a very important role on the climate and the various processes involved in the climate system (Kaufman et al., 2005; Teller and Levin, 2006; Stith et al., 2009) through their direct impact in the visibility, in the infrared (Sokolik and Toon, 1999) or the earth radiation budget (Andreae et al., 1996; Solomon, 2007) which is still poorly known. The difficulty of understanding the impact of aerosols on the Earth's radiation balance is due to the large spatial and temporal variability of their concentration and composition in the atmosphere. The mineral particles suspended in the atmosphere come from different sources and have a nature similar to the nature of the soil from which they arise (Claquin et al., 1999; Formenti et al., 2008) with a broad spectrum of particle sizes ranging between 0.01 $\mu m$ and 300 $\mu m$ (Wagener, 2008; Ryder et al., 2013). Their impact on the marine ecosystem and particularly on oceanic primary production (Duce and Tindale, 1991; Baker et al., 2003; Mills et al., 2004; Jickells et al., 2005; Mahowald et al., 2009) remains still uncertain and difficult to assess because of the composition of these particles and of physico-chemical processes affecting them ( e.g., Friese et al., 2016). Mineral dust deposition also have a negative impact on human health and are responsible for meningitis epidemics or cardiac deseases (Thomson et al., 2006; Martiny and Chiapello, 2013; Diokhane et al., 2016; Prospero et al., 2005; Griffin, 2007).

Although the transport of mineral dust across the Atlantic Ocean started to be investigated in the 1960s, it started to be studied from satellite observations since 1970s (Kaufman et al., 2005; Taghavi and Asadi, 2008). Passive sensors have the advantage of providing daily data on the state of the atmosphere with good spatial and temporal coverage. The satellite products have improved our knowledge of the source regions and dust transport pathways in recent years (Engelstaedter et al., 2006; Schepanski et al., 2007, 2009b, 2012). However, studies of their spatial and temporal variability are mainly based on indices such as the Aerosol Optical Depth (AOD) or the Aerosol Index (AI) which provide vertically integrated information on the atmospheric aerosol contents (passive space derived observations: Cakmur et al. (2001); Chiapello and Moulin (2002); Kaufman et al. (2005); Engelstaedter et al. (2006); Schepanski et al. (2009b)). Moreover AOD estimated by satellite integrates the contribution of every kind of particles and this latter estimation also depends on the altitude at which aerosols are located. Based on perturbations induced by the Rayleigh scattering in the detection of absorbing aerosols, Chiapello et al. (1999) showed that TOMS AI is more sensitive to aerosols present at high altitude than at low altitude. In other words the signal changes with the height of the aerosol plume for a given aerosol content.

Recently, the vertical structure of the Saharan Air Layer (SAL) has been analyzed from CALIPSO satellite observations. The vertical discontinuity of dust layers between land and ocean strongly impacts the atmospheric deposition rates of mineral mat-

ters (Schepanski et al., 2009a) and dust concentration at the oceanic surface which has important consequences on the primary
biological productivity of surface waters (Martin, 1992; Arístegui et al., 2009).
In boreal summer, SAL is characterized by hot, dry air, very dust-laden and is located between 10°N and 25°N (Dunion and
Marron, 2008; Tsamalis et al., 2013). This SAL is marked by very strong potential temperatures up to 40°C and a radon pres-
ence (radon-222) indicating the desert origin of air masses (Carlson and Prospero, 1972).
In winter, the SAL is characterized by the transport of dust containing chemical elements such as aliminum (Al), silicon (Si),
iron (Fe), titanium (Ti) and manganese (Mn) ( e.g., Formenti et al., 2001; Ben-Ami et al., 2010) and is located between 5°N and
10°N ( e.g., Tsamalis et al., 2013). Some of studies relating aerosols to their transport are generally a simple description of the
vertical distribution of aerosols in the SAL (Generoso et al., 2008; Liu et al., 2008; Ben-Ami et al., 2009; Braun, 2010; Yu et al.,
2010; Adams et al., 2012; Ridley et al., 2012; Yang et al., 2012) or a description of the seasonality of the SAL in connection
with large-scale dynamics (Liu et al., 2012; Tsamalis et al., 2013). However, the dust field campaigns, AMMA, SAMMUM-1
and 2, FENNEC or SALTRACE (see Table 1) of Weinzierl et al., (2016) carried out in West Africa and over the Atlantic Ocean
improved our understanding of dust-dynamics interactions. During SALTRACE, a linear depolarization ratio of particles and
a relative humidity threshold of 50% were used for differentiating different types of aerosol (Weinzierl et al., 2016). Authors
showed that sea salt aerosol were restricted to the lower layer superposed by biomass-dust mixtures. They also showed that the
altitude of the mineral dust layer decreased westward. The effects of small-scale dynamics and thermodynamics for controlling
the vertical structure of desert aerosols in coastal West Africa remain unknown, and efforts made in this direction are restricted
to very sporadic case studies (Gamo, 1996; Reid et al., 2002; Petzold et al., 2011).
In this study, in-situ and satellite observations are used to describe the seasonal time-scale of mineral dust distribution. We first
used complementary information, provided by SeaWiFS and OMI which deliver extensive (AOD) and intensive (SSA, AE)
parameters of desert aerosols, to analyse the spatial variability of the desert aerosol dust. Then we used CALIOP lidar on board
CALIPSO to investigate the vertical distribution of these desert aerosols.
We finally analyze meteorological data to explain the impact of the atmospheric variables on the seasonal cycle of the vertical
distribution of desert aerosols at the transition zone between the continent and the ocean. We conclude the present work by
summarizing all the results which are reflecting our common knowledge on mineral dust discrimination and spatio-temporal
distribution.
**2  Methodology and Data**
**2.1  AErosol RObotic NETwork (AERONET)**
We first used data of AOD from AERONET between January 2005 and December 2010. AERONET is a global network of in-
situ observations developed by the NASA Earth Observing System (NASA's EOS) (Dubovik et al., 2000). AERONET consists
of solar photometers Cimel providing measures of AOD every 15 minutes, refractive index and also allows inversions such as
particle size distribution of aerosols and single scattering albedo (SSA) at 440nm, 670nm, 870nm and 1020nm wavelengths
(Holben et al., 1998) with an accuracy of ± 0.01 (Slutsker and Kinne, 1999; Dubovik et al., 2000; Holben et al., 2001). This
uncertainty is inherent in the algorithm inversion used to retrieve aerosol characteristics. Some approximations are used in
the numerical inversion algorithm which produce errors named relative errors having a standard deviation of 0.01 (Dubovik
et al., 2000). AERONET's SSA are computed for favorable atmospheric conditions (AOD 440 nm > 0.4 and solar zenith angle
>45°) using an algorithm which performs almucantar inversions (Jethva et al., 2014). These data are used to validate remotely
sensed AOD and SSA measurements. AERONET is available under three different products: Level 1.0, 1.5 and 2.0. In this
study, we use Level 1.5 product for Cape Verde, due to a lack of sufficient Level 2 data, for this station and Level 2.0 for the
other stations. Concerning the temporal resolution of AERONET observations, we compute a "daily" mean based upon data
collected between 10am and 3pm in order to use observations collected during the same time window as satellite overpass. We
then use this 10am-3pm daily averages to compute monthly 10am-3pm AOD.

## 2.2 Sea-viewing Wide Field-of-view Sensor (SeaWiFS)

We then used DeepBlue-SeaWiFS monthly mean AOD at 550 nm and AE products derived from SeaWiFS developed by NASA
to study ocean color. SeaWiFS measures the solar radiation reflected at the top of the atmosphere in the wavelengths 412 nm,
443 nm, 490 nm, 510 nm, 555 nm, 670 nm, 765 nm and 865 nm. Satellite measurements carried out between October 1997 and
December 2010 (Jamet et al., 2004; Hsu et al., 2012) have a value of signal-to-noise and uncertainty of 2%-3% for the different
spectral bands (for details see Eplee et al. (2007); Franz et al. (2007); Eplee Jr et al. (2011)). In this paper, we use the Level 3
version 4 products (Bettenhausen and Team, 2013) for years 2005 to 2010. The SeaWiFS AOD provided at 550 nm is available
both over the land and over the ocean (Hsu et al., 2004; Sayer et al., 2012). The products used here are land-ocean estimates
generated and made available to the scientific community by NASA (Wang et al., 2000). Regarding the contribution of the
aerosols types in the AOD, the studies of Dubovik et al. (2002), Schepanski et al. (2009b) or Tegen et al. (2013) suggested that
the coarse mode fraction of mineral dust dominates the atmospheric mixture when AE values, associated with AOD values
greater than or equal to 0.3, are below 0.7. Here, we consider aerosols optical thickness larger than 0.2 when the Ångström
Exponent is lower than 0.7 (figure 4) to monitor the evolution of coarse (upper and lower bounds respectively) and fine (lower
and upper bounds) modes of mineral aerosols.

## 2.3 Ozone Monitoring Instrument (OMI)

OMI is a passive sensor on board the Aura satellite launched on 15 July 2004 by NASA's EOS Aura space-craft which released
its first observations in October 2004. Like all satellites in the A-Train constellation, OMI scans the entire Earth in 14 to 15
orbits with a nadir ground pixel spatial resolution of $13 \times 24$ km$^2$ (Jethva et al., 2014). In addition to the ozone content in the
atmosphere OMI provides information on aerosols, clouds, gases (NO2, SO2, HCHO, BrO, and OClO) and irradiance in the ul-
traviolet (Levelt et al., 2006). We use Aura/OMI SSA at 500 nm taken from https://ozoneaq.gsfc.nasa.gov/data/lance-browse/,
the OMAERUV Level 3 Collection 003 aerosol product processed in March 2012 with a spatial resolution of 1° x 1° to quantify
the scattering of the aerosol types with passive sensors. The OMAERUV algorithm assigns flag to each pixel which carries
information on the quality of the retrieval (Jethva et al., 2014).
The SSA represents the ratio (ranging between 0 and 1) of scattering coefficient to extinction coefficient and provides infor-
mation about the absorbing properties of the aerosols. SSA of 0.9 indicates that 90% of the total extinction of solar light is
caused by scattering and 10% by absorption effects (Jethva et al., 2014). This parameter depends on the wavelength, size and
the complex refractive index of particles (Léon et al., 2009). The closer this value is to one the more desert aerosols dominate
(Johnson et al., 2008; Léon et al., 2009; Ialongo et al., 2010; Malavelle, 2011).
OMI data were interpolated on the grid of SeaWiFS data to superimpose the products (AOD and SSA).

## 2.4 Cloud Aerosol Lidar and Infrared Pathfinder Satellite Observations (CALIPSO)

The first polarization lidar in space so-called CALIPSO is a sun-synchronous satellite developed by NASA as part of the Earth
System Science Pathfinder program (ESSP) and launched on April 28, 2006 (Winker et al., 2007; Hunt et al., 2009) in order
to provide a global coverage of the vertical distribution of the properties of clouds and aerosols (Winker, 2003). The CALIOP
lidar (LIght Detection and Ranging) onboard CALIPSO acquires vertical profiles of the atmosphere at 30 m resolution in the
lower layers (from the two orthogonal components that result from depolarization of a signal backscattered laser at 532 nm and
vertical profiles of a total laser at 1064 nm signal backscattered at nadir). The final level-2 product is reduced to a uniform res-
olution calculated from averaging and/or interpolating different resolutions for generating intermediate products (Winker et al.,
2006). We use the Vertical Feature Mask (VFM; stage 1 Version 3) for which the processing algorithm is described in CALIOP
Algorithm Theoretical Basis, Part 3: Scene Algorithms Classification (Liu et al., 2005). VFM allows to separate aerosols from
clouds but also the desert aerosols from other types of aerosols (Omar et al., 2009). This methodology of discrimination by
CALIOP of aerosol types gives results close to another method of distinction between mineral dust made from inversions (SSA
and AE) of AERONET level 2 products (Mielonen et al., 2009). The mix of layers of desert aerosol and other types of aerosols
(i.e. biomass burning) is very rare (Chou et al., 2008; Heese and Wiegner, 2008) in our region of interest. During the dry season,
mineral aerosols are observed in the atmospheric surface layer ranging 0.5 to 1 km while the aerosol emitted through biomass
burning are carried to higher levels up to 5 km altitude (Cavalieri et al., 2010). Nevertheless, classification errors are possible
for low values of the Mineral Dust Occurrence Frequency (MDOF) and at frontal zones between layers of different substances
(Adams et al., 2012). For this reason we only consider here the values of MDOF above 10%. Our method for determining the
mineral dust by a calculation of the MDOF is equivalent to Adams et al. (2012) and follows the equation:

$$p(x,y,z) = \frac{\sum\limits_{n=0}^{N} p(x+n,y,z)}{\sum\limits_{n=0}^{N} s(x+n,y,z)} \qquad \forall \quad x,y,z \tag{1}$$


where p is the frequency of occurrence of dust at a grid point, s the total number of valid satellite passing the same grid point and N the total number of grid points. The Occurrences in the longitude (x) are summed and normalized by the total valid satellite passes in a given longitudinal range (35°W-20°E). Data were gridded with a near-uniform horizontal resolution of 0.5° x 0.5° and a vertical resolution of 30 m for 290 vertical levels between 0.5 and 8.2 km above sea level. The CALIOP lidar on CALIPSO (also in the A-train) has a 90 m instantaneous footprint which is smeared to 333 m in the along track direction by orbital motion over the lidar pulse duration. All satellites of the A-train constellation, such as CALIPSO, fly in a sun-synchronous orbit with a 16 days coverage cycle consisting of 233 orbits separated by 1.54 degrees longitude or about 172 km at the equator. Each satellite completes 14.55 orbits per day with a separation of 24.7 degrees longitude between each successive orbit at the equator. These CALIPSO orbits are controlled to cover the same ground with cross-track errors of less than ±10 km (Winker et al., 2007). This drastically reduces the spatial coverage of the satellite. Consequently, we use a mesh of 0.5° longitude to cover the area between 10°W-24°W and 12°N-21°N. The choice of this band of latitude is driven by one of the objectives of the paper which is to study the transition of aerosol distribution between the continent and the ocean. Dust occurrences are averaged over latitudes 12°N to 21°N and are then smoothed over 30 points longitudinal running mean and 50 points vertical running mean.

## 3 Results

### 3.1 Horizontal dust distribution

SeaWiFS AOD (estimated at wavelength 550 nm) represents an average value of the optical Depth of the atmosphere. It has first been compared to the monthly AOD given by AERONET photometers (given at the wavelength 675 nm and interpolated at 550 nm) by calculating the correlation between the two measurements at different selected stations (Fig. 1). Our choice focused on the stations Banizoumbou (2.665°E-13.541°N), Agoufou (1.479°W-15.345°N), M'bour (16.959°W-14.394°N) and Capo Verde (22.935°W-16.733°N) to assess the quality of satellite information obtained accross the land-ocean continuum. A very good correlation is calculated between SeaWiFS and in-situ measurement given by the photometer at Banizoumbou (R=0.97; Fig. 1a). The photometer Cimel at Agoufou (Mali) also shows a very good correlation with SeaWiFS (R=0.87; Fig. 1b). The correlation between the two measures is equal to 0.81 at the shore in M'bour (Fig. 1c). It is close to the one in Capo Verde (R=0.83; Fig. 1d). All these correlation values of AOD are significant at 95% using a student statistical test. The regression for M'bour site is not as good as for the other sites. This site is located at the shore at the interface between land and sea and the satellite algorithm retrieval is not the same over the land and over the ocean. We also studied the structure of the cloud of points between the two datasets to assess the quality of the satellite measurements as a function of the aerosol concentration. The regression line obtained by the least squares method shows a linear relationship between satellite and in-situ monthly mean measurements of AOD at the selected stations.

The horizontal transport of desert aerosols can be followed by considering the key and complementary parameters that distinguish them. To better characterize the desert aerosols, we combined AOD (SeaWiFS) with SSA (OMI) to specify the contribution of the latter compared to other types of aerosols in the atmosphere. A threshold of 0.90 in monthly averaged SSA is used

to define regions dominated by desert aerosols. This value is chosen in agreement with the threshold value given in previous
studies (Léon et al., 2009; Malavelle, 2011; Jethva et al., 2014). This method allowed us to define the Sahelo-Saharan region
as the one which is the most influenced by dust plumes composed of desert aerosols throughout the year (between 12°N and
21°N; Fig. 3).
The comparison of the daily SSA of Aura/OMI versus AERONET is achieved to validate satellite SSA which provides a better
spatio-temporal coverage of our region of interest. OMI SSA retrievals are taken between 10am and 3pm time range which
cover AERONET measurements. As emphasized by Jethva et al. (2014), this comparison is done at the original wavelengths
of each independent measurement (388 nm for OMI and 440 nm for AERONET) in order to avoid uncertainties induced by the
interpolation at other wavelengths. Good correlations are retrieved between the two datasets at the different ground stations in
West Africa for the period 2005-2010 within root mean square (RMS) difference of 0.03 in the selected region (Fig. 2). Glob-
ally, the OMAERUV SSA is well correlated with ground measurements. The correlation at all selected sites for this study is
significant. The agreement between the two inversions is better over the continent (Banizoumbou station, r=0.47 and Agoufou
station: r=0.50) and at the shore of West Africa (M'bour station: r=0.66) than over the ocean (Capo Verde station: r=0.30).
The discrepancy between the AERONET SSA retrievals over continent (Banizoumbou and Agoufou) and at the shore of West
Africa (M'bour) was already found by Johnson and Osborne (2011) during GERBILS campaign over West Africa. These au-
thors suggested that a lack of sampling may affect the results. Their results are in agreement with our results which show 449
retrievals in Banizoumbou againts 178 retrievals in M'bour site.
Figure 3 shows a seasonal distribution of the AOD which superimposed onto SSA in West Africa region. Both, large AOD
and strong SSA indicate that mineral dust is the dominant component of aerosols in the atmosphere. In winter, the main dust
source in West Africa, the Bodélé depression, is showed in Figure 3a with AOD larger than 0.5 and SSA larger than 0.9 around
17°N-18°E. This most persistent dust hot spot is activated all along the year and provides a maximum dust emission in spring
(Fig. 3b), in agreement with Engelstaedter and Washington (2007). In summer, the intense surface heating from solar radiation
(Heat Low) induces the development of a near-surface thermal low pressure system over northern Mali, southern Algeria, and
eastern Mauritania (Lavaysse et al., 2009; Messager et al., 2010) and controls the dry convective processes which contribute to
about 35% of the global dust budget (Engelstaedter and Washington, 2007). Over Northwestward Sahara region (Fig. 3c), the
AOD is larger than 0.5 and SSA is stronger than 0.9, both variables indicate the main hot spot of mineral dust source in West
Africa in summer which has already been shown by Engelstaedter and Washington (2007).
Figures 3 and  4 show that horizontal monthly average of AOD is stronger above the continent than over the ocean throughout
the year. The weakest AOD is given for winter months (DJF for December-January-February) with a mean value of 0.33 ±
0.07 (standard deviation). At this season, the SSA values are higher in the northeast tropical Atlantic than on the West African
continent with a SSA maximum reaching 0.95. This indicates a stronger contribution of dust over the ocean than over the con-
tinent in the latitude range 12°N-21°N. Note that sources of dust aerosols are also indicated by high SSA values north of 21°N.
The air masses advection in the lower atmosphere (925 hPa) follows a NorthEast-SouthWest direction in winter (Fig. 6a), dust
coming from the NorthWest of Mauritania is partially seen over the continent (in AOD and SSA) and its main signature should
be seen over the ocean. In spring (MAM for March-April-May), the increase of the monthly mean AOD compared to winter is
indicated by a stronger mean value (0.50 ± 0.08). The mean optical depth indicates that the dust sources are becoming more
active with an atmosphere more charged than in winter. The coarse mode dominates in the mixed atmosphere boundary layer
over the continent with lower values of AE less than 0.7 (not shown). Nevertheless, the reflectance properties of aerosols (given
by the SSA) is higher over the ocean than over the continent and vary weakly compared to winter.
In summer (JJA for June-Jully-August), the maximum mean AOD is 0.52 ± 0.05. AOD values are associated with higher SSA
above 0.96. It indicates that aerosols are clearly dominated by desert dust in boreal summer. At this season, important quantity
of dust can be lifted up and vertically transported in the upper atmosphere by convective systems and near-surface convergence
(Engelstaedter and Washington, 2007).
In autumn (SON for September-October-November), the monthly mean AOD is 0.34 ± 0.05. AOD is decreased compared to
spring but the SSA values are much higher than in spring despite the fact that uplift occurrences are larger in spring than in fall
in West Africa (Marticorena et al., 2010; Diokhane et al., 2016).
Changes of AOD and SSA are seen at the transition between the continent and the ocean (Fig. 4). Understanding these changes
requires a thorough analysis of the vertical distribution of dust during transportation from east to west in North Africa.

## 3.2 Vertical dust distribution

The vertical distribution of desert aerosol indicates a strong presence of dust concentrations between the surface and 6 km in
agreement with the results of Léon et al. (2009) who studied the vertical distribution of dust in the North-East Tropical Atlantic
(Fig. 5).
In DJF, desert aerosols are mainly concentrated in the atmospheric boundary layer (ABL) between the surface and 2 km
(Fig. 5a) both over the continent and the ocean. At this season, we also noted a homogeneous dust aerosol transition between
Western Africa and the Eastern part of the Atlantic Ocean.
In MAM, there is an elevation of the SAL with a maximum altitude of 5 km on the continent and between 4 and 5 km above
the ocean (Fig. 5b). The MDOF over 50% above the continent shows that dust emissions are much greater than in winter. The
ABL is developed vertically to reach up to 5 km of altitude. It results in an atmospheric layer well mixed between the surface
and 5 km of altitude above the continent (10°W-15°W). Above the Ocean we see a detachment of the SAL from the ocean
surface which occurs at the coast (around 18°W).
JJA is the busiest season of the year in terms of dust rising in the northern hemisphere of Africa. It is characterized by the de-
velopment of density currents that intensify the mobilization of terrigenous aerosols (e.g., Bou Karam et al., 2008; Schepanski
et al., 2009b, Fig. 5c). Unlike DJF, we note a clear separation of the dust layer above the Eastern Atlantic Ocean where dusts
are confined between 1 and 6 km altitude.
In SON, dust emissions decrease in intensity compared to JJA but the detachment from the surface of the ocean remains clear
at the coast although less marked than in JJA (Fig. 5d). According to Adams et al. (2012), the heart of the SAL is located about
5 km above sea level in SON, whereas Liu et al. (2012) shows a maximum altitude of 4 km.

## 4 Discussion

### 4.1 Seasonal variability

The desert aerosols in the band of latitude 12°N-21°N are mainly emitted in the Saharan and Sahelian regions. Emissions and transport processes are mainly controlled by meteorological variables (Brooks and Legrand, 2000; Joseph, 1999). Schepanski et al. (2009b) found that over the Sahara sources of dust emissions are less active in winter than during summer season. The southward migration of the ITCZ and the subsiding branch of the Hadley cell over the dry convection can also prevents the deep vertical distribution of aerosols in north Africa (Lavaysse et al., 2009). The maximum altitude of this distribution is 3 km above the continent and 2 km at the West African coast in agreement with the studies of Léon et al. (2009) and Vuolo et al. (2009). Compared to other seasons, DJF show an important role played by the shallower atmospheric layers on the dust transported from source regions located in the Northwestern part of Mauritania and more generaly in the West African coastal region (Fig. 6a). This high occurrence is shown by the inter-seasonal variability derived from NCEP Reanalysis. Figure 6 highlights that the Northwest region of Mauritania has the highest standard deviation of horizontal wind intensity between 18°N-24°N and that wind is very intense in winter compared to the other seasons (Fig. 6a). Hence this region represents an important sand source in winter as mentioned by previous studies (Bertrand et al., 1979; Ozer, 2000; Tulet et al., 2008; Laurent et al., 2008; Mokhtari, 2012; Hourdin et al., 2015).

Unlike winter, as showed in Figure 5c, dust are concentrated between the higher layers of the ABL, from one to 5-6 km (Gamo, 1996), in response to intense convective mechanisms that are more common in the region at this season (Cuesta et al., 2009). Indeed, the summer solar heating drives the development of the Saharan boundary layer which reaches up to 6 km while the convergence of hot dry air (Harmattan) from the Sahara with fresh moist air (monsoon) from the ocean generates intense convective cells which are responsible for the suspension of large amounts of dust which will be distributed in the ABL. Transport is also growing between 3 and 4 km above the ocean with a MDOF greater than 70%, i.e. more than 30% higher than that observed in DJF. This sharp increase of MDOF from DJF to JJA is in agreement with the results of Schepanski et al. (2009b) who estimated an increase of more than 20% of the activity of dust sources in summer compared to winter in West Africa in the observations of Meteosat Second Generation (MSG) Spinning Enhanced Visible and Infrared Imager (SEVIRI). In summer, atmospheric dynamics raise large dust particles that are settling down much closer to the source regions than the rest of the year (Shao, 2000). However, their reflectivity of solar radiation becomes larger and reaches a maximum value indicated by a SSA of 0.97 (Fig. 4c).

In autumn, SSA values are comparable to spring values but these high values are not due to high reflectance of desert aerosols like in spring because the southern migration of the Inter-Tropical Convergence Zone (ITCZ) reduces the activity of convective systems and causes a reduction of dust emissions shown by a decreasing of the AOD (Fig. 4d). These high SSA values can be attributed to atmospheric conditions seen through the relative humidity which is much higher than in spring (Fig. 7d). Indeed, OMI measures the atmospheric properties of the aerosols which are known to be hygroscopic (Jethva et al., 2014).

## 4.2 Continent-Ocean transition

To better understand the factors responsible for the high variability of the vertical transition of desert aerosols from the continent to the ocean, we placed ourselves at a coastal point (Dakar) to study the variation of meteorological variables and their potential influence on the distribution of aerosols. Seasonality of vertical distributions of winds, relative humidity and potential temperature from radiosounding conducted at the weather station (GOOY) of Dakar (at West African shore) are shown in Figure 7.

In DJF, continental winds are very strong at the near-surface with a maximum of 25 m/s at 500 m (Fig. 7a). The north-east direction of the winds in the first thousand meters explains the homogeneity of the vertical distribution of dust from the continent towards the ocean. This north-east wind applies to all West Africa at the surface (Fig 6a). Their intensity also explain the strong values of MDOF (up to 50%) observed by CALIOP in wintertime above the continent. Between 1 and 2 km height, winds weaken and change direction (south to south-east) while MDOF observed by satellite decreases (Fig. 5a). Between 2 and 5 km height, the winds turn to the southwest and west. These dust-depleted air masses of oceanic origin are wetter than from the land, and limit the development of the ABL. The air masses of continental origin are located between the surface and 2000 m height (Fig. 7a). In Figure. 7a, the relative humidity is around 20% (between 500 and 2000 m) and it corresponds to a very dry air mass of Saharan origin. Between 2 and 5 km the potential temperature indicates a stable atmospheric layer. This season is associated with an intermediate AOD value which decreases from 15°E to 10°W. SSA reflects mineral dust properties across its westward transportation (>0.9) but is higher by 0.2 over the ocean than the continent. We believe it could reflect the transport of dust emitted along the coastline which is only partly taken into account in dust properties derived from the continent.

Compared to the DJF situation, MAM near-surface winds (Fig. 7b) are still intense with a maximum of 25 m/s at 500 m height and are from the east. They are associated with MDOF above 50% in the ABL around 14°W. Surface winds (Fig 6b) shows the near-surface convergence of northward and southward flows along 16°N which is associated with a well-mixed distribution of dust in the first 5 km of the atmosphere (Fig 5b) and higher AOD values than in winter (Fig 4). There is an inversion of easterly winds between 1 and 3 km and a second southerly wind peak (15 m/s) appears between 3 and 4 km. It corresponds to the dust layer (SAL) detected by CALIOP. The vertical profile of potential temperature indicates a stable thick layer, well mixed between the surface and 3 km (Fig. 7b). Beyond this altitude there is a stable stratification of the atmosphere indicated also by the potential temperature. Between 3 and 5 km height, the air masses coming from the South to the South-Southwest are also of oceanic origin and their interaction with a more consistent amount of dust than in winter could explain the better marked transition between the ocean and the continent in terms of SSA (increase) and AOD (decrease) for this season (Fig. 4b). Indeed, in general, increasing the relative humidity is likely to increase the SSA and size hygroscopic aerosols with dry to wet passage inducing a larger diameter even when humidity is below the saturation level (Hervo, 2013; Howell et al., 2006).

In JJA, surface winds (0-1 km) decrease and are from the West to the Southwest (West African Monsoon) (Fig. 7c). This corresponds to lower values of MDOF (Fig. 5c) but to relative humidity values well above DJF or MAM (Fig. 7). Reid et al. (2002) presented a conceptual model of Saharan dust transport in the middle troposphere describing an evolution of relative

humidity profile in agreement with the observations made in Dakar. These authors describe a moistening of the surface layers
due to monsoon flow which penetrates up to 1.5 km above this layer. Figure 6c shows deep intrusion of air masses coming
from the Golf of Guinea which brings humidity into the continent. The dry convection taking place over the continent favors
the vertical transport of dust to high altitudes (Engelstaedter and Washington, 2007).
Between 2 and 6 km, winds are from the East and above 15 m/s. These wind velocity maxima reach 25 m/s in the range 3.5-5
km and are associated to the African Easterly Jet (AEJ) (Wu et al., 2009; Lafore et al., 2011). The co-localization of the AEJ
and the SAL between 2 and 5 km height (Fig. 5c and Fig. 7c) causes the westward SAL transport by AEJ in summer (Karyam-
pudi et al., 1999). These strong winds correspond to the layer of dust detected by satellite at this altitude (Fig. 5c). Above the
continent, the mesoscale features associated with the convergence between Harmattan and the West African Monsoon at the
ITCZ cause strong updrafts that allow lifting and transport of dust particles throughout the air column (Tulet et al., 2008). The
dynamics of the monsoon described by the conceptual scheme of mechanisms controlling the dust vertical redistribution in
Cuesta et al. (2009) explain the wide occurrence of dust found between 2 and 5 km rather than at the surface. During transport
from North Africa to the Atlantic Ocean, very large amounts of coarse dust (Fig. 4c) are deposited along the path with a rapid
change in the size distribution of aerosols near the west African coast (Ryder et al., 2013). The changes of the aerosol size and
properties will impact the climate system (Huneeus et al., 2011; Mahowald et al., 2014). McConnell et al. (2008) suggested that
the variation in the aerosol profiles over the ocean have an impact on the radiative effect, a statement confirmed by Highwood
et al. (2003) who showed that the radiative effect of mineral dust is correlated with the altitude of the dust layer.
The signing of the SAL is evidenced by relative dryness of the atmosphere (Dulac et al., 2001) between 1.5 and 5 km (Fig. 7c).
At this altitude, the vertical profile of potential temperature indicates Saharan origin of air masses with temperatures between
35°C and 45°C (Carlson and Prospero, 1972). The wind direction (east) given in Figure 7c between 1.5 and 5 km altitude con-
firms the origin of the Saharan air masses. The presence of dust in the SAL causes both warming and drying of the atmosphere
between 1.5 and 5 km and a cooling below this layer (Tulet et al., 2008).
In SON, winds are weak and from the East at the surface (Figs. 7d and 6d). Between 1 and 5 km, it is increasing but is less
intense than in JJA between 3 and 5 km and it is associated with a decrease of the MDOF (Fig. 5d). The moisture profile in
SON (Fig. 7d) is close to that of JJA, but has a more humid atmosphere in the layer between 1.5 and 5 km where maximum
relative humidity of the year occurs (60%; Fig. 7d). The analysis of the vertical distribution of thermodynamic variables like
relative humidity, potential temperature and wind measured at the Dakar weather station shows that the thermodynamical con-
ditions control the dust vertical distribution as well as the depth of the dust layer depending on the season. This analysis also
explains the unintuitive differences between spring, when the low values of SSA are associated with a strong AOD, and autumn
characterized by high values of SSA associated with low AOD values.

## 5  Conclusions

Studies of processes involved in the vertical distribution of aerosols at the transition between continent and ocean are very rare.
Here, we took advantage of a weather station ideally located on the main pathway of desert aerosols from Northern Africa

(Léon et al., 2009; Marticorena et al., 2010; Mortier et al., 2016) to explain the effect of meteorological variables on this transition in a region of primary importance worldwide. The interaction of air masses of oceanic origin with dust aerosols are crucial for understanding their fate ( e.g., Friese et al., 2016). This study constitutes the first attempt to relate the seasonal dynamic of the atmosphere and the vertical distribution of dust aerosol in this region and provides the first dynamical explanation of a counterintuitive deposition pattern over the Atlantic ocean. Indeed, it explains the role of the local atmospheric circulation in driving a higher AOD and dust content in summer over west Africa in phase with dust deposition in Barbades islands but in opposition with Cape Verde islands where deposition is more intense in winter (Chiapello et al., 1995).

We have studied the seasonal variability of the distribution of desert aerosols in West Africa (continental and oceanic) from their optical and physical properties. First of all we have been able to show a good estimate of physical properties (AOD and SSA) of aerosols by satellite when compared with AERONET ground measurements on the mainland, the coast and the ocean. Space observations then allowed us to show the predominant presence of Saharan dust in the atmosphere north of 12°N throughout the year and an additional significant contribution of sandy sources from the Mauritanian coast in winter. The MDOF indicates a change in the vertical distribution of dust at the transition between the continent and the ocean, the largest differences occurring in spring and summer seasons. In DJF, the ABL is shallow ($\sim$ 1km) and strong winds from North-East transport the dust in a dry atmosphere from the continent to the ocean continuously. This surface layer is superimposed by a stable atmospheric layer which inhibits the vertical development of this surface layer rich in dust aerosols. The decrease from east to west of the AOD requires material deposition during the transit. In summer dry convection located north of 10°N and associated with structures that develop at the Inter-Tropical discontinuity (ITD) distribute dust up to 6 km height and create a thicker AOD. Above 6 km altitude over the Saharan-Sahel areas, the vertical distribution of dust is blocked by the strong subsiding branch of the Hadley cell (Lavaysse et al., 2009). In the lower layers, the westward oceanic moistly entries which are opposite to the higher eastward winds generate very different distributions above the continent and the ocean. On the mainland, the dust is dominated by coarse mode and have a homogeneous vertical distribution while above the ocean, lower layers are poor in dust and are superimposed by the SAL which is highly enriched. The SSA remains constant at this transition. MAM and SON represent transition periods. For the vertical dust distribution, MAM is being closer to the summer situation.

Future modeling experiments should bring further insights into ocean-atmosphere processes involved in explaining this transition and the dust deposition along this pathway. It also seems that a more tailored approach to ocean-atmosphere interactions including higher frequencies of variability and notably the diurnal cycle is needed to make more apparent the role of local circulation on the vertical distribution of aerosols in coastal areas.

*Acknowledgements.* We would like to thank the IRD-BMBF AWA project and the international joint laboratory ECLAIRS for supporting and promoting our research activities. We thank the Institute of Research for Development for funding this PhD. We also thank ICARE for the online availability of the CALIPSO aerosol products at http://www.icare.univ-lille1.fr/archive. NCEP Reanalysis data were found online by the http://www.esrl.noaa.gov/psd/data/gridded/data.ncep.reanalysis.pressure.html, and the PIs and NASA for online AERONET data set which can be obtained from http://aeronet.gsfc.nasa.gov/. OMI aerosol products were downloaded at http://disc.gsfc.nasa.gov/gesNews/giovanni_3_end_of_service?instance_id=omil2g&selectedMap=Blue%2520Marble&. We are very grateful to B. Marticorena and I. Chia-

384 pello for very fruitful discussions. We are finally very grateful to the two anonymous referees for very informative comments which have

385 greatly improved the quality of this study.

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

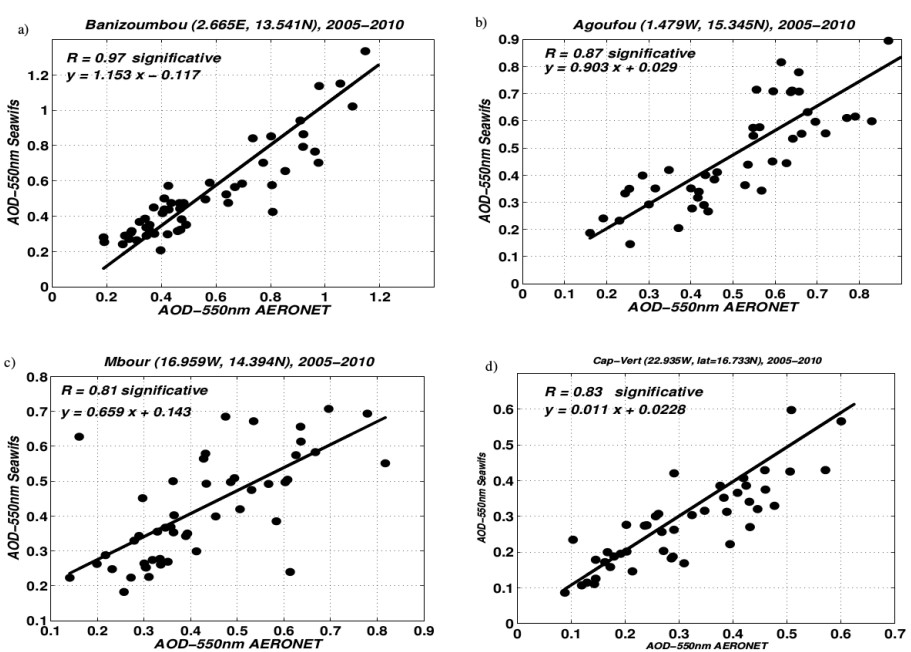

**Figure 1.** Comparison of monthly mean aerosol optical depth (AOD) between SeaWiFS (550 nm) and ground measurements from AERONET (675 nm) from January 2005 to December 2010. This comparison is done at the following stations : a) Banizoumbou (53 points), b) Agoufou (47 points), c) M'bour (50 points) and d) Cape verde (47 points). The red solid line represents the regression between both dataset

.

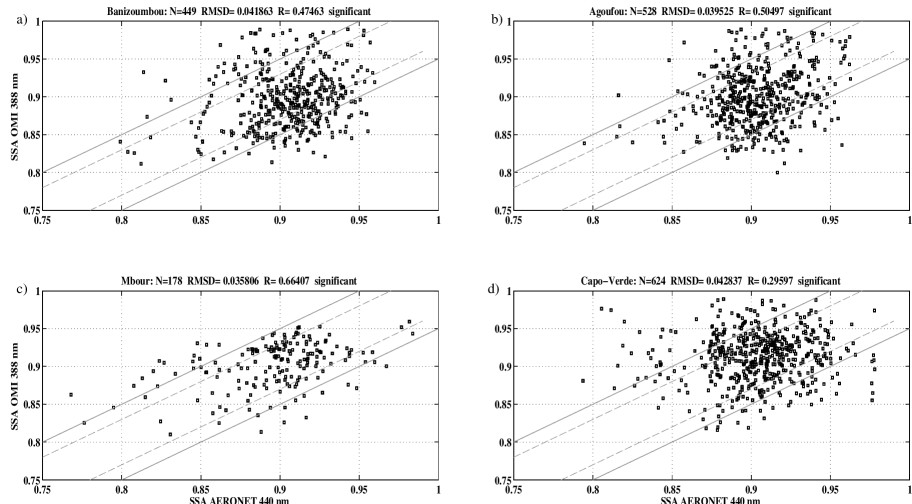

**Figure 2.** OMAERUV SSA at 388 nm wavelength as a function of AERONET SSA at 440 nm at a) Banizoumbou (2.66°E,13.54°N; 449 retrievals); b) Agoufou (1.47°W, 15.34°N; 528 retrievals); c) Mbour (16.95°W, 14.39°N; 178 retrievals) and d) Capo Verde (22.93°W, 16.73°N; 624 retrievals). The solid lines indicate the domain where the two retrievals agree with each other within 0.03 and the dashed lines indicate agreement within 0.05.

.

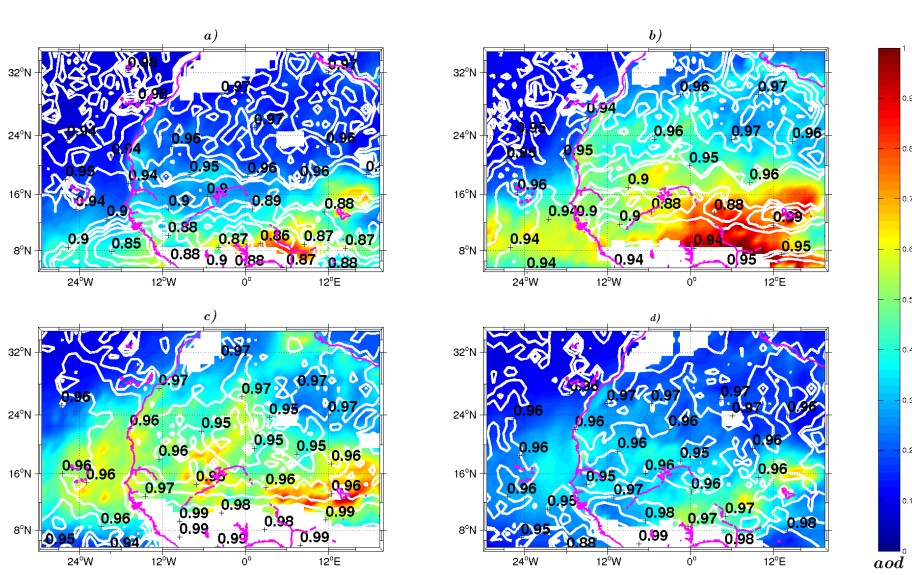

**Figure 3.** Seasonal distribution of aerosol optical depth (average between 2005 and 2010) at 550 nm wavelength (colours) from SeaWiFS for a) DJF; b) MAM; c) JJA and d) SON. SSA from OMI is superimposed with white contour lines.

.

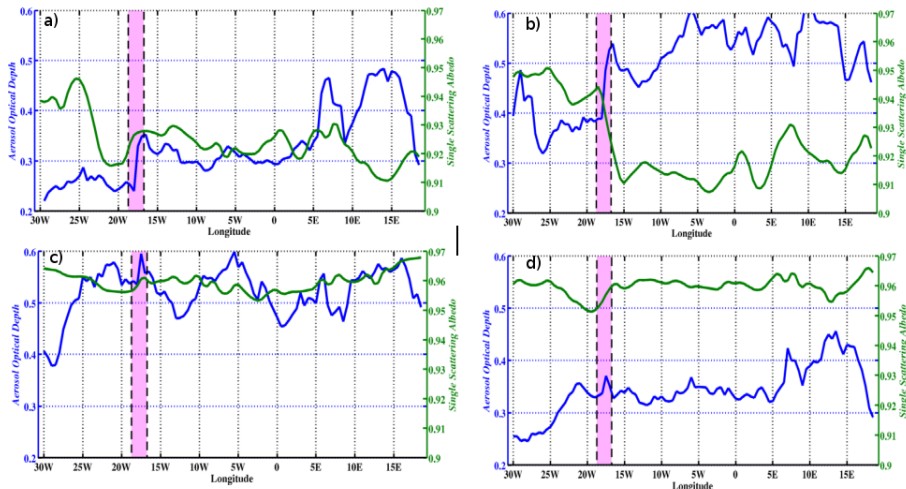

**Figure 4.** Seasonal SeaWiFS AOD at 550 nm (bleu), Aura/OMI SSA (green) zonally averaged between 12° and 21°N and from 2005 to 2010: a) DJF; b) MAM; c) JJA; and d) SON. The black dashed lines indicate the continent-ocean transition for the latitude range 12°-21°N.

.

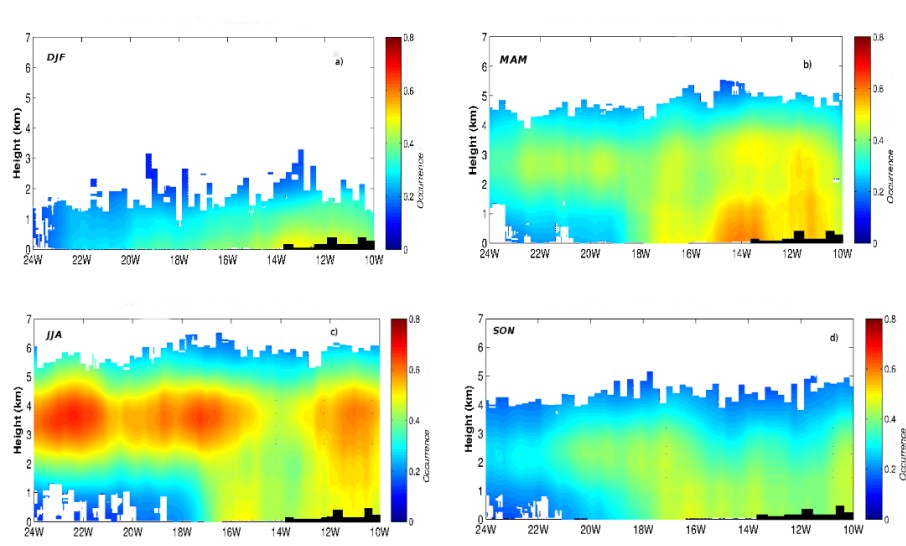

**Figure 5.** CALIOP daytime seasonal vertical distribution of the frequency of mineral dust aerosol occurrence zonally averaged between 12° and 21°N over the period 2007-2013: a) DJF; b) MAM; c) JJA; and d) SON.

.

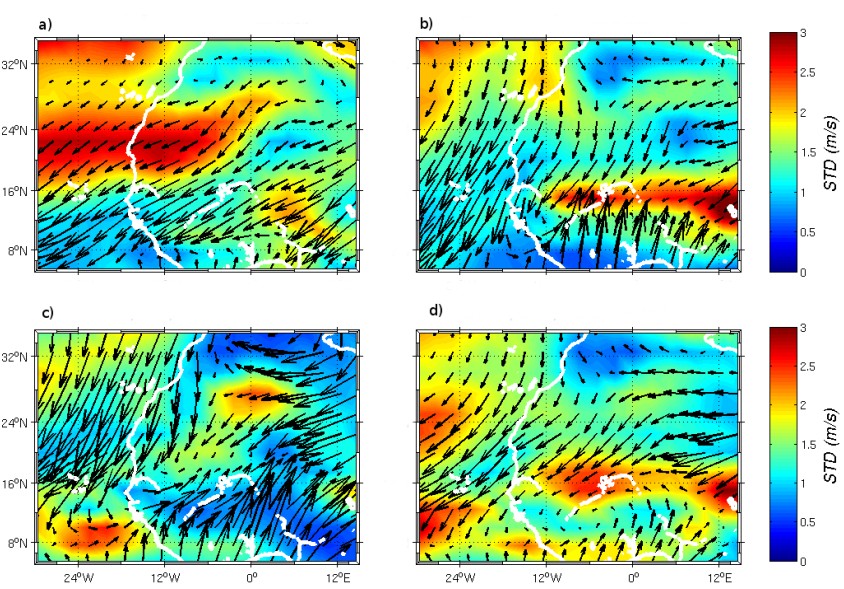

**Figure 6.** Seasonal mean zonal wind field at 925 hPa over West Africa from NCEP Reanalysis between 2000 and 2012: a) DJF; b) MAM; c) JJA; and d) SON. The vectors show wind direction while colors indicate the standard deviation of wind velocity (m.s$^{-1}$).

.

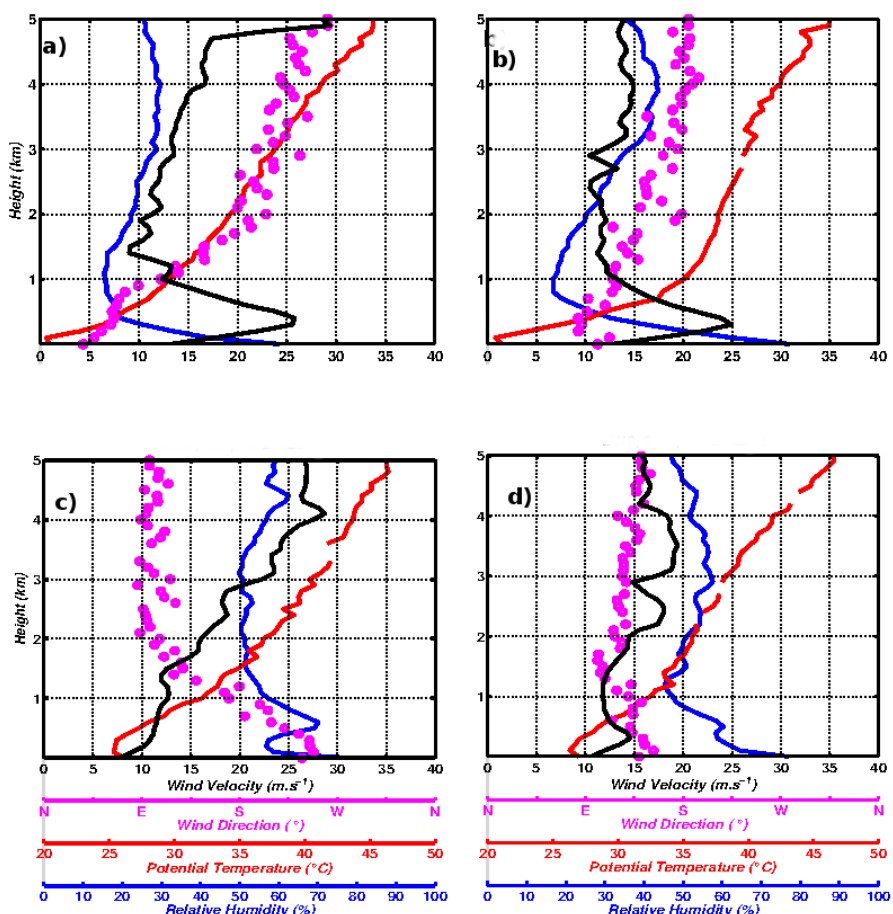

**Figure 7.** Mean seasonal vertical profiles of wind velocity (black line), wind direction (pink dots), potential temperature (red line) and relative humidity (blue line) at Dakar weather station (14.73°N, 17.51°W) for a) DJF; b) MAM; c) JJA; and d) SON. Observations correspond to weather balloon launched daily at 12UTC for years 2012 to 2014.

.