# Peer review of "Seasonal cycle of desert aerosols in West Africa : analysis of the coastal transition with passive and active sensors"

_Atmospheric Chemistry and Physics, 2016_

## Referee Comment (RC1) · Anonymous Referee #2 · 17 Jan 2017

General comments

This paper presents an interesting study of the mineral dust aerosol distribution in Western Africa and Eastern Atlantic Ocean. The study is based on the analysis of satellite aerosol products from SeaWifs, OMI and CALIPSO. The authors focus on the seasonal variability of the dust optical properties retrieved over the land and over the ocean. A second part deals with the seasonal variation of the altitude of the dust layer based on CALIOP lidar soundings and meteorological data.

I found that the main interesting point of the paper is Figure (4), which depicts the variation of the advection of mineral dust, can affect the single scattering albedo. However the explanations given by the authors are rather confusing, e.g. L177 "stronger

contribution of dust over the ocean than over the continent". This statement is counter-intuitive because dust sources are located over the continent and AOD over the continent is also higher. As well in the conclusion L324, "MAM is being closer to the summer situation": this is not the case for the SSA for which we observe a gradient between land and ocean while the vertical distribution is similar. You must clarify this point. Although it could be interesting to use a different zonal area for summer and winter because the dust transport follows a E-W direction during summer while it is NE-SW during winter as depict by Figure (3).

The paper requires several improvements before publication.

Specific comments

- L44. Clarify this sentence and add relevant references. Explain how the AOD retrieval depends on vertical extent of the dust layer.

- L44. Provide information on the quality level (level 1.5 or level 2) and temporal resolution (daily mean or temporal window around satellite overpass). It appears later in the text that you have used monthly means.

- L65. AE is an optical parameter. Extensive (AOD) and intensive (SSA, AE) parameters are more appropriate.

- L78. Improve the description of uncertainties on aerosol parameters.

- L125. Clarify what is MDOF and what is p(x,y,z). Please refer to Adams et al (2012 and clearly define equation (1) and explain all the terms.

- Figure 1 and related text starting L145. Compare AOD for the same wavelength. You can interpolate the sun photometer AOD at 500 nm from AOD at 440 and 675 nm. Comment the regression coefficients you have obtained. In particular, explain why the regression for M'Bour site is significantly different from the others.

- L165. Correct sentence. The http link must be in the data description section.

- L167 and Figure (2). It is unclear which wavelength you have used for the comparison. Please rewrite Figure (2) caption and avoid unnecessary information on site location. Why did you use daily data rather than monthly data as for the AOD validation?

- Figure 3. It is not possible to read the SSA contour lines. Provide an additional figure with SSA regional pattern.

- L230. How do you use the AE? Please clarify and clearly state this in the data and method section.

- L232. Rewrite sentence and defined correctly which layer you are talking about.

- L240. Explain the link between gravitational settlement and SSA. This whole paragraph is unclear. However it is of highly importance to get your point on the link between the dynamic of the dust transport and the optical properties. Consider also revising L298.

Technical comments:

- Avoid use of "desertic aerosol". Prefer desert aerosol or better mineral dust.

- L42: correct sentence between brackets

- L125: Adam must be outside brackets

---

## Referee Comment (RC2) · Anonymous Referee #1 · 25 Feb 2017

In this paper, the authors present an analysis of the seasonal evolution of desert aerosol properties (AOD, SSA in particular) along a longitudinal band covering a part of Sahel and of the Eastern Atlantic Ocean. They use a combination of passive and active space-borne remote sensing observations, together with meteorological data and numerical weather prediction analyses.

In spite of some merit, the paper contains too many inaccuracies. The authors do not really have a solid knowledge the dynamics/thermodynamics features of the WAM system impacting the atmospheric boundary layer over the Sahara, the dust emission and transport over the region. I have made numerous comments along these lines in the following. No mention is made to previous projects/campaign that have taken

place in the area and contributed to advance knowledge on dust-dynamics interactions: GERBILS, FENNEC, SAMUM. Some AMMA results are discussed.

I also think that the authors should discuss also to what extent the change in aerosol properties at land-sea transition has an impact on air-sea interactions or even the radiative budget in the region. . .

For the reasons above, my recommendation is "major revision". The authors need to address the lacks that are highlighted above and in the subsequent comments, before the paper can be considered as suitable for publication in ACP. I have not done the job of editing the manuscript at this stage.

Specific comments

Abstract

NCEP, CALIOP, OMI and SeaWIFS have to be defined

Introduction

L21-25: not only America. . . Transport pathways also include EU and the Mediterranean, depending on the season. Please correct.

L31: even larger particles have been found close to source regions: see the work of Ryder et al. during FENNEC

L37-38: no, the Observatory in Barbados set up by J. Prospero goes back to the 1960s

L39-40: also cite the work of Shepanski et al. who have used the high temporal resolution of SEVIRI to analyze emission hot spots location and frequency

L43-44: not true. MODIS or MISR allow identifying dust using SSA, deep blue is almost exclusively a dust product. . .

L46-47: what signal. . . be more specific

L48: the SAL is defined L 52

L48-49: The vertical disconnection of dust layers between land and ocean: I do not understand this. The SAL is an emanation of the Saharan ABL which is undercut by the low level flow from the Atlantic. . . this flow penetrates over the continent during the day.. so that the disconnection is not necessarily appearing at the land-sea transition..

L55-56: are those elements part of the composition of dust? Otherwise where do they come from?

L57: not true, see the recent BAMS paper on the SALTRACE campaigns

L67: CALIOP and CALIPSO need to be defined.

Section 2: methodology and data What stations did you use? They should be listed here. . . Are you using level 2 data only?

Section 3 : results

L171-173: can you explain why the SSA-related correlations with ARONET stations are so low compared to the AOD-related correlations which are quite good. . .? Is this link to the threshold of 0.9 that you have selected? Changing this threshold to a higher value may improve the correlations. . . Also, it is unclear why the SSA correlations a better near the coast (M'Bour) where I would expect greater mixing of dust with other particles) than over the continent where dust should be present almost exclusively. . .Also you are saying that the SS correlation in Cape Verde is better than over the continent, which is not true based on the numbers given in the text: 0.3 in Cape Verde Vs 0.47 and 0.5 inland.

Figure 3: To what dust hot spot are the largest AOD values observed related to? From your map this looks to be the Aïr region? What is happening there? In your domain you are not including the Bodélé depression, arguably the largest dust source in the world, why? Why are the largest AOD values observed in MAM? What are the dynamical processes related to these emissions?

L187: unclear what is meant here by "largest dust particles are mobilized and raised

above the continent by convective systems".. Are you referring to dust mobilization at the leading edge of cold pools? This is a very efficient mechanism for sure, but only when soil moisture is low and vegetation has not grown yet, i.e. May and June over Sahel... Please be more explicit what is meant here. Do you refer to Fig. 4c of Rajot et al?

Generally speaking, I think this Figure should be better described with more insights into the processes and hot spot regions leading to the observed distribution of AOD and SSA.

L202: "ABL develops vertically to reach the level of the SAL." No! The SAL is an emanation of the SABL has explained above... the fact that the aerosol layer is detached from the surface is related to the flow from the Atlantic penetrating over the continent. One way to show that would be to add zonal winds (in the form of arrows) in the cross-section: you would see westerly winds where the there is no or littles dust in the low levels and easterly winds in the upper levels where dust is observed.

Discussion

L218: "[...] and the vertical distribution of aerosols is not supported by a favorable wind regime ascending particles." What do you mean? This is very unclear...

L222: "These West African emission zones participate actively to the transport of mineral aerosols in the near Atlantic Ocean.": this is lame, please rephrase

L233-236: It is the dry convection related to the solar heating that drives the development of the Saharan boundary layer and hence the fact that dust aerosols are seen up to 6 km or more in the summer. What is happening in the region of the ITD is marginal in this process... The dust layer overpassing the monsoon flow maybe be slightly elevated due to the cold air undercutting the warmer dust-laden air, but the monsoon flow is not deep enough to account for the change in elevation of the top of the SAL.

L241: "In summer, atmospheric dynamics raise large dust particles that are subject

to the law of universal gravitation of Newton," Tell me something that is not!! How pompous and meaningless is that?

Section 4.2: I feel this section should be better tied up with the discussion of Figures 4 , 5 and 6 as it brings essential dynamics and thermodynamics information to the reader not familiar with West African weather.

Conclusion

L320: "In summer, convection associated with structures that develop at the ITCZ distribute dust over 6 km height and create a thicker AOD." I totally disagree.. It is dry convection over the Sahara and northern Sahel that controls the height of the top of the SAL and the altitude at which the dust from eastern sources towards the west.

---

## Author Comment (AC1) · 15 Apr 2017

We would like to thank the Reviewer for carefully reading of manuscript and for numerous useful suggestions.

*1) I found that the main interesting point of the paper is Figure (4), which depicts the variation of the advection of mineral dust, can affect the single scattering albedo. However the explanations given by the authors are rather confusing, e.g. L177 "stronger contribution of dust over the ocean than over the continent".*
*This statement is counter-intuitive because dust sources are located over the continent and AOD over the continent is also higher.*

We agree that this sentence is counter-intuitive. AOD is indeed stronger above the continent than over the ocean because the atmosphere also contains aerosols issued from biomass burning occurring south of our region of interest. As transport follows a NE-SW direction in winter, dust coming from the NW of Mauritania is partially seen over the continent (in AOD and SSA) and its main signature should be seen over the ocean. Hence, we attribute the high SSA values encountered over the Ocean to dust originated from this NW Mauritanian source which should exclusively be composed of mineral dust.
We have modified the text L225-227 (in the new version) to clarify our explanation.

*2) As well in the conclusion L324, "MAM is being closer to the summer situation": this is not the case for the SSA for which we observe a gradient between land and ocean while the vertical distribution is similar. You must clarify this point.*

L324 (in the previous version), we were referring to the vertical distribution when stating that "MAM is being closer to the summer situation". However, SSA over the ocean is indeed higher than over the continent (Figure 4). The same reason as mentioned above applies here. Over the continent, AOD is high in the southern part of our domain (Figure 3b) where aerosols from biomass burning mix with aerosol dust giving SSA values lower than 0.9 (Malavelle, 2011). As the transport is still NE-SW, SSA over the ocean still records a higher contribution of mineral dust over the ocean than the continent. This explains the higher SSA over the ocean. Comment was added in the text L391.

*3) Although it could be interesting to use a different zonal area for summer and winter because the dust transport follows a E-W direction during summer while it is NE-SW during winter as depict by Figure (3).*

Our objective was to study the vertical distribution of mineral dust and to better understand the fate of dust through the land-ocean transition. We therefore tried to prevent the influence of other aerosols coming from biomass burning because they have different optical and chemical properties and hence have a different fate. As biomass burning occurs south of 12°N in winter and spring time (Engelstaedter et al., 2007), we decided to take this latitude as southern limit. We also wanted to keep a coastline oriented as "north-south" as possible in order to be able to locate the continent-ocean transition before looking at the aerosol properties above the ocean and land. As the coastline is oriented northeast/southwest north of Cape Blanc, we

chose 21°N as northern limit. But indeed, winter and summer main transport directions are different but we believe the choice of this latitude band allows to capture most of the mineral dust signal.

*4) - L44. Clarify this sentence and add relevant references. Explain how the AOD retrieval depends on vertical extent of the dust layer.*

Based on perturbations induced by the Rayleigh scattering for the detection of absorbing aerosols, Chiapello et al., (1999) showed that TOMS AI is most sensitive to aerosols at high altitudes. We have added the a sentence to clarify this statement in the text L55-57.
The results of these authors were also related by Kaufman et al. (2005).

*5) Provide information on the quality level (level 1.5 or level 2) and temporal resolution (daily mean or temporal window around satellite overpass). It appears later in the text that you have used monthly means.*

AERONET is available under three different products: Level 1.0, 1.5 and 2.0. In this study, we have used Level 1.5 for Cape Verde and Level 2.0 for the other stations. We used Level 1.5 product for Cape Verde due to a lack of sufficient Level 2 data for this station.
Level 1.5 data are raised to Level 2.0 (quality-assured) after final calibration values are applied and manual data inspection is completed (http://aeronet.gsfc.nasa.gov/new_web/man_data.html; Smirnov et al., 2000a). At Level 1.5, the minimum aerosols optical depth ($\tau_{ai\ min}$) is identified at each wavelength ($\tau_{ai}$) and for each station. If the difference $\tau_{ai\ -\ \tau_{ai\ min}}$ is less than the maximum of (0.05*$\tau_{ai\ min}$, 0.02) for each channel, then cloud is affected to this record. If this screening removes all but one point from a series then an additional criterion is applied to the spectral channels. If the Angström parameter computed using all available channels between 440 and 870 nm is greater than -0.1, then the point is considered cloud and pointing error free.
The final post-deployment calibration values are applied to the data set for producing Level 2.0 products. The spectral channels are evaluated for data anomalies, filter degradation or other possible instrument failures. The data are also inspected for possible cloud contaminated outliers.
Concerning the temporal resolution of AERONET observations, we computed a "daily" mean based upon data collected between 10am and 3pm in order to use observations collected during the same time window as satellite overpass. We then used these 10am-3pm daily averages to compute monthly 10am-3pm AOD.
The complementary information is added to the text L100-102.

*6) L65. AE is an optical parameter. Extensive (AOD) and intensive (SSA, AE) parameters are more appropriate.*

We took this remark into account and modified the text accordingly L82.

*7) L78. Improve the description of uncertainties on aerosol parameters.*

The uncertainty we are talking about L79 (in the previous version) concerns AERONET data. This uncertainty is inherent in the algorithm inversion used to retrieve aerosol characteristics. Some approximations are used in the

numerical inversion algorithm which produce errors named relative errors having a standard deviation of 0.01 (Dubovik et al., 2000). A comment was added in the text L96-98.

*8) L125. Clarify what is MDOF and what is p(x,y,z). Please refer to Adams et al (2012 and clearly define equation (1) and explain all the terms.*

We are not sure to properly understand Reviewer's comment. MDOF is the Mineral Dust Occurrence Frequency (L151). In equation (1), p(x,y,z) is equivalent to MDOF. Indeed, it is not a probability of occurrence but a frequency of occurrence. L159
We added the following explanation of equation (1) .
The Occurrences in the longitude (x) are summed and normalized by the total valid satellite passes in a given longitudinal range (35°W-20°E).
p is the resulting occurrence frequency at the grid point, s the number of valid satellite passes at the same grid point, and N the number of grid points in a specified longitudinal range. It was now clarify in the text L160-161.

*9) Figure 1 and related text starting L145. Compare AOD for the same wavelength. You can interpolate the sun photometer AOD at 500 nm from AOD at 440 and 675 nm.*
*Comment the regression coefficients you have obtained. In particular, explain why the regression for M'Bour site is significantly different from the others.*

We agree with Reviewer. We now have interpolated (L176-177) AERONET AOD at 500 nm from AOD at 440 and 675 nm (new Figure 1). The correlations between AERONET and satellite data did not change significantly, nor the slopes or intercepts of the linear regression.  L180-181
Indeed, the regression of AOD for M'bour site is not as good as for the other sites. M'bour is located at the shore at the interface between land and sea. The satellite algorithm retrieval is not the same over the land and over the ocean. As M'bour experiences both oceanic and continental influences (notably through wind diurnal cycles), we believe the AOD retrieval at the shore is more complexe than in land (Banizoumbou and Agoufou) or at sea (Cape verde). L183-185

*10) L165. Correct sentence. The http link must be in the data description section.*

This sentence  was corrected accordingly L197
The link was now moved in the data and methodology section. L125

*11) L167 and Figure (2). It is unclear which wavelength you have used for the comparison.*
*Please rewrite Figure (2) caption and avoid unnecessary information on site location. Why did you use daily data rather than monthly data as for the AOD validation ?*

The right  wavelength which used in this work is now written in the text L199
Figure 2 caption was wrong, we have now corrected it. The right wavelength is indicated on the x- and y- axes in the figure. We also removed unnecessary information about site location.

For the evaluation of the performance of satellite SSA retrievals we indeed used daily AERONET SSA using observations between 10am and 3pm. We used these daily observations to obtain significant correlations and robust regressions (see also our response to comment number 15 of Reviewer #2).

*11) Figure 3. It is not possible to read the SSA contour lines. Provide an additional figure with SSA regional pattern.*

We believe the superimposition of SSA contour levels onto AOD is better since they both represent mineral dust characteristics. We have improved the quality of our Figure 3. We hope this new Figure 3 will indeed be easier to read.

*12) L232. Rewrite sentence and defined correctly which layer you are talking about.*

According to Reviewer's remark, we reformulated the sentence as follow:
"Unlike winter, dust are concentrated between the higher layers of the ABL, from one to 5-6 km (Fig. 5C; Gamo, 1996), in response to intense convective mechanisms that are more common in the region at this season (Cuesta et al., 2009)." L282-283

*13) L230. How do you use the AE ? Please clarify and clearly state this in the data and method section.*

We agree with Reviewer. We now have clarified our use of AE in the data and method section.
Here we use aerosols optical thickness larger than 0.2 when Ångström Exponent is lower than 0.7 (see L116). This methodology is based on AOD and AE to characterise mineral dust and has already been used by Ben-Ami et al., (2010) and Drame et al., (2015).

*14) L240. Explain the link between gravitational settlement and SSA. This whole paragraph is unclear. However it is of highly importance to get your point on the link between the dynamic of the dust transport and the optical properties. Consider also revising L298.*
We did not intend to link the settlement of large particles to SSA properties. SSA remains high and roughly constant throughout the continent and over the ocean. We believe summer AOD in northern Africa is largely dominated by mineral dust which could explain the high SSA values encountered at this season. L294
We understand that the paragraph starting L298 (in the previous version) could be misleading. We therefore clarify this point in the manuscript. L364

*Technical comments:*
*- Avoid use of "desertic aerosol". Prefer desert aerosol or better mineral dust*
We have changed "desertic aerosol" expression accordingly throughout the manuscript (in the title and in the whole text)
*- L42: correct sentence between brackets*
Corrected accordingly (L53)
*- L125: Adam must be outside brackets*
Corrected accordingly (L154)

---

## Author Comment (AC2)

We would like to thank the Reviewer for having carefully read of the manuscript and for having made numerous useful suggestions.

*1) In this paper, the authors present an analysis of the seasonal evolution of desert aerosol properties (AOD, SSA in particular) along a longitudinal band covering a part of Sahel and of the Eastern Atlantic Ocean. They use a combination of passive and active space-borne remote sensing observations, together with meteorological data and numerical weather prediction analyses. In spite of some merit, the paper contains too many inaccuracies. The authors do not really have a solid knowledge the dynamics/thermodynamics features of the WAM system impacting the atmospheric boundary layer over the Sahara, the dust emission and transport over the region. I have made numerous comments along these lines in the following.*

*No mention is made to previous projects/campaign that have taken place in the area and contributed to advance knowledge on dust-dynamics interactions: GERBILS, FENNEC, SAMUM. Some AMMA results are discussed.*

We agree that very interesting knowledge was collected during the field campaigns carried out in the region. The FENNEC and the African Monsoon Multidisciplinary Analysis (AMMA) campaigns were used in Ryder et al., (2013), Cuesta et al., (2009) and Marticorena et al., (2010)  which were cited (in the previous version). Saharan Mineral Dust Experiment (SAMUM) was also indirectly mentioned  through  Petzold et al., (2011). Please see the previous version (L287,L233,L190 L62 etc).
Puerto Rico Dust Experiment (PRIDE) campaign was also used in Reid et al. (2002) which was also cited L62 and L274 (in the previous version).
To this respect, we believe we took these observation campaigns into account to highlight the gain of knowledge that they implied, although we indeed didn't mention explicitly name them.
However, we acknowledge the interest of results issued from other field studies and we added a reference to Weinzierl et al. (2016) who published results from the SALTRACE experiment and McConnell et al.'s (2008) associated to the Dust Outflow and Deposition to the Ocean (DODO) project. L72-L75 etc, L349
The Saharan Dust Experiment (SHADE) and the Geostationary Earth Radiation Budget Intercomparison of Longwave and Shortwave radiation (GERBILS) were indirectly cited  through  Highwood et al., (2003)  and Johnson and Osborne (2011), respectively. See the manuscript (L351 and L207)

*2) I also think that the authors should discuss also to what extent the change in aerosol properties at land-sea transition has an impact on air-sea interactions or even the radiative budget in the region.*

We agree with the Reviewer that the changes in aerosol properties will have an impact on the surface heating and hence on the heat fluxes at the interface and probably on winds at a regional scale. We added sentences to highlight this aspect but note that the paper aims at describing the seasonal changes of aerosol distribution and properties at the land-sea transition and does not pretend to answer about the climate effect of these changes.

Liao and Seinfeld (1998) or Kok (2011) showed that the radiative interactions are sensitive to the size of the mineral dust aerosols and their optical properties. These aerosols properties change during their transport along which they are affected by sedimentation and other processes.

We mentioned in the manuscript "During transport from North Africa to the Atlantic Ocean, very large amounts of coarse dust (Fig. 4c of the manuscript) are deposited along the path with a rapid change in the size distribution of aerosols near the west African coast (Ryder et al., 2013)" (L286-287 in the previous version). The changes of the aerosol size and properties during their transport will indeed impact the climate system (Huneeus et al., 2011, Mahowald et al., 2014).

McConnell et al. (2008) suggested that the variation in the aerosol profiles over the ocean has an impact on the radiative effect. Indeed, Highwood et al. (2003) showed that the radiative effect of the mineral dust aerosols is correlated with the altitude of the dust layer. Our Figure 5 clearly shows the seasonal variation of the vertical distribution. Hence, we added a sentence to discuss the fact that this seasonality will impact the radiative budget.

The modification of the Single Scattering Albedo across the land-sea transition also affect significantly the radiative budget. Having a higher SSA over the ocean than over the continent implies a stronger dust-induced cooling of the ocean relative to the continent. This differential cooling impacts the temperature gradients across the land-sea transition and hence might affect the wind. We added this discussion in the manuscript (L348-351)

*3) Abstract*
*NCEP, CALIOP, OMI and SeaWIFS have to be defined.*

The acronyms NCEP (National Center for Environmental Prediction), CALIPSO (Cloud Aerosol Lidar and Infrared Pathfinder Satellite Observations), CALIOP (Cloud-Aerosol Lidar with Orthogonal Polarization), OMI (Ozone Monitoring Instrument) and SeaWiFS (Sea-Viewing Wide Field-of-View Sensor) are now defined in the abstract.

*4) L21-25: not only America. . . Transport pathways also include EU and the Mediterranean, depending on the season. Please correct.*

This paragraph was corrected as follow (L27-30):

The mineral dust aerosols emitted from the Sahara desert can be transporter over long distances in the atmosphere and can be detected as far as Americas (Prospero et al., 1981; Swap et al., 1992; Formenti et al., 2001; Kaufman et al., 2005; Ansmann et al., 2009b; Ben–Ami et al., 2010), Mediterranean region (Bergametti et al., 1989; Moulin et al., 1998; Hamonou et al., 1999; Gobbi et al., 2000; Ansmann et al., 2003; Papayannis et al., 2008, Schmechtig et al., 2010) and Asia (Ganor and Mamane, 1982; Israelevich et al., (2003), Ganor et al., 2010).

But here, the study of dust transport focuses on the main corridor of their transport Westward Africa (Formenti et al., 2001).

*5) L31: even larger particles have been found close to source regions: see the work of Ryder et al. during FENNEC*

We added Ryder's (2013) reference and corrected the sentence as follow (L37):
The mineral particles suspended in the atmosphere come from different sources and have a nature similar to the nature of the soil from which they arise (Claquin et al., 1999, Formenti et al 2008) with a broad spectrum of particle sizes ranging between 0.01 μm  and 300  μm (Wagener, 2008; Ryder et al., 2013).

*6) L37-38: no, the Observatory in Barbados set up by J. Prospero goes back to the 1960s*

In this sentence, we refer to the study of the transport from satellite information. In order to clarify the eventual amibiguity, we reformulate the sentence as follow:
Although the transport of mineral dust across the Atlantic Ocean started to be investigated in the 60s, it started to be studied from satellite observations in the  1970s (Kaufman et al., 2005, Taghavi et al., 2008). L44-46

*7) L39-40: also cite the work of Shepanski et al. who have used the high temporal resolution of SEVIRI to analyze emission hot spots location and frequency*

We agree with the Reviewer. This sentence was modified by adding Schepanski et al., (2007, 2009 and 2012) studies (L48). These authors have done a very interested work on the mineral dust based on SEVERI sensor to analyze emission hot spots location and frequency. Their work was cited in the previous version  (L43,L49,L206,L227,L238).

*8) L43-44: not true. MODIS or MISR allow identifying dust using SSA deep blue is almost exclusively a dust product. . .*

We agree with Reviewer's remark and hence decided to remove the following sentence L52
"These satellite products also present some limitations since they are unable to differentiate aerosols and particularly those from desertic origin."

*9) L46-47: what signal. . . be more specific*

We clarified that by writing L55-57:
"Moreover AOD estimated by satellite integrates the contribution of every kind of particles and this latter estimation also depends on the altitude at which aerosols are located. Based on perturbations induced by the Rayleigh scattering in the detection of absorbing aerosols, Chiapello et al. (1999) showed that TOMS AI is more sensitive to aerosols present at high altitude than at low altitude. In other words the signal changes with the height of the aerosol plume for a given aerosol content.

*10) L48: the SAL is defined L52*
*It was now corrected accordingly L59*

*11) L48-49: The vertical disconnection of dust layers between land and ocean: I do not understand this. The SAL is an emanation of the Saharan ABL which is undercut by the low level flow from the Atlantic. . . this flow penetrates over the continent during the day.. so that the disconnection is not necessarily appearing at the land-sea transition..*

First, we want to mention that we used disconnection instead of discontinuity and we apologize for this misuse (L60).
This discontinuity has already been mentionedin Chiapello et al. (1995) and Tsamalis et al., (2013) who used bothground measurements and satellite observations in their study. The satellite observation from CALIPSO was used in this work and results show a clear seasonal discontinuity of the dust layer at the land-ocean transition (see Figure 5 of the manuscript).

*12) L55-56: are those elements part of the composition of dust? Otherwise where do they come from ?*

Indeed, we did not relate theses elements to dust in this sentence. We reformulated the sentence as follow L66-67:
"In winter, the SAL is characterized by the transport of dust containing chemical elements such as aliminum (Al), silicon (Si), iron (Fe), titanium
(Ti) and manganese (Mn) (e.g., Formenti et al., (2001); Ben-Ami et al., (2010); and is located between 5°N and 10°N (e.g., Tsamalis et al., (2013))."

*13) L57: not true, see the recent BAMS paper on the SALTRACE campaigns*
*"The studies relating aerosols to their transport are generally a simple description of the vertical distribution of aerosols in the SAL (Generoso et al., 2008; Liu et al., 2008; Ben-Ami et al., 2009; Braun, 2010; Yu et al., 2010; Adams et al., 2012; Ridley et al., 2012; Yang et al., 2012) or a description of the seasonality of the SAL in connection with large-scale dynamics (Liu et al., 2012; Tsamalis et al., 2013)."*

We would like to mention that our work was initiated before the availability of the mentioned reference. However we took into account Reviewer's suggestion by adding a sentence to the former one L71-76:
"Some of studies relating aerosols to their transport are generally a simple description of the vertical distribution of aerosols in the SAL (Generoso et al., 2008; Liu et al., 2008; Ben-Ami et al., 2009; Braun, 2010; Yu et al., 2010; Adams et al., 2012; Ridley et al., 2012; Yang et al., 2012) or a description of the seasonality of the SAL in connection with large-scale dynamics (Liu et al., 2012; Tsamalis et al., 2013). However, the dust field campaigns, AMMA, SAMMUM-1 and 2, FENNEC or SALTRACE (see Table 1 of Weinzierl et al., (2016) carried out in West Africa and over the Atlantic Ocean improved our understanding of dust-dynamics interactions. During SALTRACE, a linear depolarization ratio of particles and a relative humidity threshold of 50% were used for differentiating different types of aerosol (Weinzierl et al., 2016). Authors showed that sea salt aerosol were restricted to the lower layer

superposed by biomass-dust mixtures. They also showed that the altitude of the mineral dust layer decreased westward."

*14) L67: CALIOP and CALIPSO need to be defined.*

*All acronyms have now been defined.*

*15) Section 2: methodology and data What stations did you use? They should be listed here. . . Are you using level 2 data only ?*

This following paragraph was added in the methodology and data section to clarify the data which have been used in this study L100-104.
"These data are used to validate remotely sensed AOD and SSA measurements. AERONET is available under three different products: Level 1.0, 1.5 and 2.0. In this study, we use Level 1.5 product for Cape Verde, due to a lack of sufficient Level 2 data for this station and Level 2.0 for the other stations (Banizoumbou, Agoufou and M'bour). Concerning the temporal resolution of AERONET observations, we compute a "daily" mean based upon data collected between 10am and 3pm in order to use observations collected during the same time window as satellite overpass. We then use these 10am-3pm daily averages to compute monthly 10am-3pm AOD."

*16) L171-173: can you explain why the SSA-related correlations with ARONET stations are so low compared to the AOD-related correlations which are quite good. . .? Is this link to the threshold of 0.9 that you have selected? Changing this threshold to a higher value may improve the correlations. . . Also, it is unclear why the SSA correlations a better near the coast (M'Bour) where I would expect greater mixing of dust with other particles) than over the continent where dust should be present almost exclusively. .*

We agree with the Reviewer. A comment was added in the text (L205-209).
The SSA-related correlations with AERONET stations are low compared to the AOD-related correlations for the same stations. Both quantities (AOD and SSA) are computed from AOD and its absorption since SSA=[AOD-absorption(AOD)]/AOD. This uncertainty can then be related to the fact that AOD used for AOD or for SSA comes from two different sensors (SeaWIFS and OMI respectively) and processing. As seen in the calculation of SSA, its computation uses two quantities (AOD and its absorption) and hence introduces different sources of errors.
We don't believe that this uncertainty is related to our threshold of 0.9. This value has already been used in previous studies in this region and it seems to be a good threshold for differentiating dust aerosols from biomass burning aerosols (Léon et al., 2009, Jethva et al., 2014).
During the GERBILS campaign over West Africa, Johnson and Osborne (2011) showed that there was no obvious relationship between the observed SSA and the geographic location of the measurement.
The better correlation of SSA at M'bour station could be due to different reasons.
It can be due to the strong contribution of dust transport from the northwest flow which advects important dust amounts from Mauritanian sources in winter (Figure 6a).
The OMAERUV algorithms of OMI retrievals could provide another

explanation. Indeed, cloud contamination of thin cirrus in the OMI footprint can cause the overestimation of SSA retrieval (Jethva et al., 2014).

During the GERBILS campaign over West Africa, Johnson and Osborne (2011) found that AERONET SSA retrievals show significantly lower SSAs at Banizoumbou than Dakar (averaging 0.91 and 0.94, respectively). These authors suggested that this discrepancy could be due to the smaller amount of sampling in Dakar. In agreement with the later authors, we have fewer sampling in our case at M'Bour than over the continent (see caption of Figure 2).

A comment was added in the text to clarify that discrepancy on SSA correlations (L205-L209).

*Also you are saying that the SSA correlation in Cape Verde is better than over the continent, which is not true based on the numbers given in the text: 0.3 in Cape Verde Vs 0.47 and 0.5 in land.*

We apologise for the wrong formulation, this sentence has been reformulated as follow (L203-204):

"The agreement between the two inversions is better over the continent (Banizoumbou station, r=0.47 and Agoufou station: r=0.50) and at the shore of West Africa (M'bour station: r=0.66) than over the ocean (Capo Verde station: r=0.30)".

*17) Figure 3: To what dust hot spot are the largest AOD values observed related to? From your map this looks to be the Aïr region? What is happening there? In your domain you are not including the Bodélé depression, arguably the largest dust source in the world, why? Why are the largest AOD values observed in MAM? What are the dynamical processes related to these emissions?*

We do not completely understand Reviewer's point. The Bodélé depression is located at 17°N-18°E and hence is part of our domain (12°N-21°N and 35°W-20°E).

Concerning the largest AOD encountered in MAM, Figure 3b shows a large AOD south of 21°N at this season. These large AOD values result from a mixing of aerosols of dust and biomass burning origin. The presence of biomass burning aerosols is indicated by SSA values lower than 0.90 south of 15°N. Léon et al., (2009) were also surprised to find low values of SSA at Mbour coastal station (~14°N) in March during which strong dust events occur. On the other hand, the presence of dust north of 15°N is also highlighted. It is in agreement with the map of dust emissions showed by (Weinzierl et al. 2016; Fig.5). Figure 6b also shows the convergence of cold flow from ocean and the warm flow from North Africa which results in highly variable winds which is at play in the dust emission in this region. All together, biomass burning and dust emissions occurring at the end of the dry season are responsible for the largest AOD values encountered in MAM.

Comment was added in the text (L211-214).

*18) L187: unclear what is meant here by "largest dust particles are mobilized*

*and raised above the continent by convective systems".. Are you referring to dust mobilization at the leading edge of cold pools? This is a very efficient mechanism for sure, but only when soil moisture is low and vegetation has not grown yet, i.e. May and June over Sahel. . . Please be more explicit what is meant here. Do you refer to Fig. 4c of Rajot et al?*

We understand that this sentence was unclear. It was replaced (L234-235) by: In boreal summer important quantity of dust can be lifted up and vertically transported in the upper atmosphere by convective systems.

*19) Generally speaking, I think this Figure should be better described with more insights into the processes and hot spot regions leading to the observed distribution of AOD and SSA.*

We agree with the Reviewer. Hence, the following comment was added to the text (L210-219):

Figure 3 shows the seasonal distribution of SSA superimposed on AOD in West Africa. Together, high AOD and high SSA indicate the dominance of dust aerosol in the atmosphere. In winter, the main dust source in West Africa, i.e. the Bodélé depression, is depicted with AOD greater than 0.5 and SSA higher than 0.9 around 16°N-18°E (Fig.3a). This persistent dust hot spot is activated all along the year and exhibits a maximum of dust emission in spring (Fig.3b), in agreement with Engelstaedter et al. (2007). In summer, the intense surface heating from solar radiation (Heat Low) induces the development of a near-surface thermal low pressure system over northern Mali, southern Algeria, and eastern Mauritania (Lavaysse et al., 2009; Messager et al., 2010) and controls the dry convective processes which contribute to about 35% of the global dust budget (Engelstaedter et al., 2007; Fig.3c). Over the Northwest Saharan region (16N-24N,0-12W), AOD is higher than 0.5 and SSA is around 0.95 indicating a large area of dust emission at this season already mentioned by Engelstaedter et al., (2007).

*20) L202: "ABL develops vertically to reach the level of the SAL." No! The SAL is an emanation of the SABL has explained above. . . the fact that the aerosol layer is detached from the surface is related to the flow from the Atlantic penetrating over the continent. One way to show that would be to add zonal winds (in the form of arrows) in the cross-section: you would see westerly winds where the there is no or littles dust in the low levels and easterly winds in the upper levels where dust is observed.*

We agree with the Reviewer that the SAL is an emanation of the SABL. This sentence was modified as follow (L250):
"The ABL is developed vertically to reach up to 5 km of altitude."
Indeed, we could have added the zonal winds as arrows in the cross-section in Figure 5. Kaufman et al. (2005) showed good correlations between NCEP zonal wind and AOD in West Africa all along the year. Here, we have used NCEP horizontal wind at 925 hPa (Figure 6c) and radiosounding data over Dakar to discuss the effect of the atmospheric circulation on the vertical aerosol distribution. Both figure 6 and 7 show that the flow below 1 km comes from the Atlantic Ocean while above 1 km. It comes from the West African continent.

*21) L218: "[. . .] and the vertical distribution of aerosols is not supported by a favorable wind regime ascending particles." What do you mean? This is very unclear. . .*

Referring to the large-scale dynamics, the deep vertical dust transport is mainly controlled by the Inter-tropical Convergence Zone (ITCZ). The ITCZ exhibits a clear seasonal latitudinal migration and is located around 5°N (e.g. Sultan et al., 2000) in winter. In terms of general circulation, the subsiding branch of the Hadley cell blocks the vertical transport of dust by dry convection in the region  (Lavaysse et al., 2009).
Engelstaedter et al., (2007) showed that in North Africa the strong surface convergence, associated dry convection and increased vertical wind velocity create conditions that favor dust emission and transport into higher altitudes.
We reformulated the sentence to clarify our point (L265-268):
"(Schepanski et al, 2007) found that over the Sahara sources of dust emissions are less active in winter than in summer season. The southward migration of the ITCZ and the subsiding branch of the Hadley cell also prevents the vertical distribution of aerosols to develop (Lavaysse et al., 2009)."

*22) L222: "These West African emission zones participate actively to the transport of mineral aerosols in the near Atlantic Ocean.": this is lame, please rephrase*

We agree with the Reviewer that this sentence does not bring any insight. Hence we decided to remove (L272) this sentence and the next one which does not bring information either.

*23) L233-236: It is the dry convection related to the solar heating that drives the development of the Saharan boundary layer and hence the fact that dust aerosols are seen up to 6 km or more in the summer. What is happening in the region of the ITD is marginal in this process. . . The dust layer overpassing the monsoon flow maybe be slightly elevated due to the cold air undercutting the warmer dust-laden air, but the monsoon flow is not deep enough to account for the change in elevation of the top of the SAL.*

We are agree with the Reviewer that this sentence was misleading.
Indeed, on one side, the solar heating drives the development of the Saharan boundary layer and on the other side, the low pressures located over North Africa around 23°N (Lavassy et al., 2009) induces the increase of dust activity (Choobari et al., 2014). We amended this sentence as follow (L284-289):
"Indeed, the summer solar heating drives the development of the Saharan boundary layer which reaches up to 6 km while the convergence of hot dry air (Harmattan) from the Sahara with fresh moist air (monsoon) from the ocean generates intense convective cells which are responsible for the suspension of large amounts of dust which will be distributed in the ABL."

*24) L241: "In summer, atmospheric dynamics raise large dust particles that are subject to the law of universal gravitation of Newton," Tell me something that is not!! How pompous and meaningless is that?*

This sentence was modified as follow (L294-295):
"In summer, atmospheric dynamics raise large dust particles that are settling down much closer to the source regions than the rest of the year (Shao, 2000). "

*25) Section 4.2: I feel this section should be better tied up with the discussion of Figures 4, 5 and 6 as it brings essential dynamics and thermodynamics information to the reader not familiar with West African weather.*

We have chosen to structure the manuscript to discussing the seasonality before describing the influence of dynamics and thermodynamics on the vertical distribution at the interface between land and ocean. Hence we took into account Reviewer's comment and tried to tie up better section 4.2 to compile the information contained in Figures 4, 5 and 6 but we believe the final scheme of what is observed in the different Figures is well synthesized in the conclusion section that is made for it.

*26) L320: "In summer, convection associated with structures that develop at the ITCZ distribute dust over 6 km height and create a thicker AOD." I totally disagree.. It is dry convection over the Sahara and northern Sahel that controls the height of the top of the SAL and the altitude at which the dust from eastern sources towards the west.*

This sentence was modified as follow (L384-387):
In summer dry convection located north of 10°N and associated with structures that develop at the Inter-Tropical discontinuity (ITD) distribute dust up to 6 km height and create a thicker AOD. Above 6 km altitude over the Saharan-Sahel areas, the vertical distribution of dust is blocked by the strong subsiding branch of the Hadley cell (Lavaysse et al., 2009).